# p38- and ERK-MAPK Signalling Modulate Developmental Neurotoxicity of Nickel and Vanadium in the *Caenorhabditis elegans* Model

Omamuyovwi M. Ijomone [1,2,3,*], Ann-Kathrin Weishaupt [1,4], Vivien Michaelis [1], Olayemi K. Ijomone [1,3] and Julia Bornhorst [1,4,*]

1   Food Chemistry, Faculty of Mathematics and Natural Sciences, University of Wuppertal, 42119 Wuppertal, Germany; weishaupt@uni-wuppertal.de (A.-K.W.); michaelis@uni-wuppertal.de (V.M.); oolaibi@unimed.edu.ng (O.K.I.)
2   Department of Human Anatomy, School of Basic Medical Sciences, Federal University of Technology Akure, Akure 340252, Nigeria
3   Laboratory for Experimental and Translational Neurobiology, University of Medical Sciences, Ondo 351101, Nigeria
4   TraceAge-DFG Research Unit on Interactions of Essential Trace Elements in Healthy and Diseased Elderly (FOR 2558), Berlin-Potsdam-Jena-Wuppertal, 14558 Nuthetal, Germany
*   Correspondence: omijomone@unimed.edu.ng (O.M.I.); bornhorst@uni-wuppertal.de (J.B.)

**Abstract:** Nickel (Ni) and vanadium (V) are characteristic heavy metal constituents of many crude oil blends in Sub-Saharan Africa, and we have previously demonstrated their neurotoxic impact. However, molecular mechanisms driving Ni and V neurotoxicity are still being elucidated. The p38- and ERKs-MAPK pathways, which are mostly known for their involvement in human immune and inflammatory signalling, have been shown to influence an array of neurodevelopmental processes. In the present study, we attempt to elucidate the role of p38- and ERK-MAPK in neurotoxicity after early life exposures to Ni and V using the *Caenorhabditis elegans* model. Synchronized larvae stage-1 (L1) worms were treated with varying concentrations of Ni and V singly or in combination for 1 h. Our results show Ni induces lethality in *C. elegans* even at very low concentrations, while much higher V concentrations are required to induce lethality. Furthermore, we identified that loss-of-function of *pmk-1* and *pmk-3*, which are both homologous to human p38-α (MAPK14), is differentially affected by Ni and V exposures. Also, all exposure scenarios triggered significant developmental delays in both *wild-type* and mutant strains. We also see increased mitochondrial-derived reactive oxygen species following Ni and V exposures in *wild-type* worms with differential responses in the mutant strains. Additionally, we observed alterations in dopamine and serotonin levels after metal exposures, particularly in the *pmk-1* strain. In conclusion, both Ni and V induce lethality, developmental delays, and mitochondrial-derived ROS in worms, with V requiring a much higher concentration. Further, the results suggest the p38- and ERK-MAPK signalling pathways may modulate Ni and V neurodevelopmental toxicity, potentially affecting mitochondrial health, metal bioavailability, and neurotransmitter levels.

**Keywords:** metals; MAPK; *C. elegans*; dopamine; serotonin; nervous system

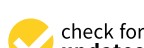



## 1. Introduction

Environmental and occupational exposures to heavy metals are an increasing health concern all over the globe. Heavy metals are implicated in a variety of neurological diseases, including Alzheimer's disease, Parkinson's disease, cerebrovascular disorders (including stroke), and neurodevelopmental disorders, amongst others [1,2]. Remarkably, heavy metals are amongst the most common pollutants released from crude oil exploratory activities [3,4]. Crude oil production accounts for a major part of the revenue of many Sub-Saharan African governments, particularly Nigeria, which is the largest oil producing country in Africa. However, the negative impact of crude oil on human health remains

hugely concerning. Worrisomely, countries within the Sub-Saharan African region are especially prone to spillage, and further saddled with the ill-capacity to implement appropriate control measures to prevent socio-economic and health setbacks [5,6].

Nickel (Ni) and vanadium (V) are characteristic heavy metal constituents of several crude oil blends in Nigeria, and other Sub-Saharan African countries [7,8]. Soil and water contamination by Ni and V due to oil exploratory activities particularly after oil spillage have been documented [3,4]. We have previously demonstrated neurotoxic impact following overexposure to Ni in rats [9,10]. Furthermore, other groups have reported neurotoxic consequences following V treatment in both mice and rats [11,12]. Neurotoxic effects of both metals reported in experimental rodents include behavioural deficits, neuronal degeneration, inflammation, apoptosis, oxidative stress, as well as mitochondrial dysfunction. Furthermore, both metals have been shown to trigger developmental deficits in the nervous system following early life exposure [13–16]; however, molecular mechanisms driving Ni and V neurotoxicity are still being elucidated.

Mitogen-activated protein kinases (MAPKs) consists of a family of serine/threonine kinases signalling molecules that are involved in a wide variety of cellular processes. The p38 mitogen-activated protein kinases (p38), extracellular signal-regulated kinases (ERKs), and c-Jun N-terminal kinases (JNKs) subfamilies are the best characterized of the MAPKs signalling pathways. The MAPKs are activated by threonine and tyrosine phosphorylation catalysed by MAPK1 or MAPK/ERK kinases (MEKs). The MAPK1 are in turn activated by MAPK2 or MEK kinases (MEKKs), which themselves respond to several extracellular stimuli such as environmental stressors and growth factors. Thus, the MAPKs pathways contain three major successively activated protein kinases; MAPK2—MAPK1—p38-/ERK-/JNK-MAPK [17–19]. The most known function of the MAPKs is their involvement in human immune and inflammatory signalling [20–22]. However, more recent evidence has clearly demonstrated that p38- and ERK-MAPK specifically influence a wide array of neurodevelopmental processes including neuronal/glia induction, formation, survival, proliferation, and differentiation [18,23,24].

Environmental toxicants such as metals could modulate cellular processes via the MAPK pathways [18]. Therefore, in the present study we attempt to elucidate the role of p38- and ERK-MAPK in developmental toxicity from early life exposures to Ni and V. Here we have utilized the *Caenorhabditis elegans* model, a relatively easy to maintain and genetically modifiable experimental model that is hugely contributing to the understanding of human risks to environmental toxicants [25]. Further advantages include its short life span (~20 days), large brood size, and transparent body, which allows relatively fast in vivo experimental set ups. Its fully sequenced genome with over 70% homology to human genome and a completely mapped neural network [26] permits elucidation of genetic control of many neuronal processes [25]. The *C. elegans* p38- and ERK-MAPK pathways are highly homologous to their human counterparts [20]. The genes *pmk-1, pmk-2,* and *pmk-3* are the worm homologues of human p38-MAPK genes, while *mpk-1* is homologous to the human ERK-MAPK gene [17,27]. In the present study, we evaluated the effects of early life exposures to Ni and V regarding toxicity, metal bioavailability, mitochondrial-derived ROS, and neurotransmitter levels (dopamine and serotonin) in *C. elegans* with loss-of-function mutations of *pmk-1, pmk-3,* and *mpk-1* genes compared to *wild-type* worms.

## 2. Results

### 2.1. Toxicity and Developmental Delays after Ni and V Treatment

Dose–response survival curves show that Ni induces lethality in exposed *wt C. elegans* at very low concentrations with an $LD_{50}$ (lethal dose causing 50% death) of 2.52 mM (Figure 1a). On the other hand, V exposure requires much higher concentrations to induce lethality with an $LD_{50}$ of 107.30 mM (Figure 1b). A 5:1 mixture of Ni and V shows similar lethality as Ni only with $LD_{50}$ of 2.53 mM (2.11 mM Ni: 0.42 mM V) (Figure 1c). Deletion mutants of *pmk-1* show similar sensitivity exposed to Ni only and metal mixtures as *wt* worms with $LD_{50}$ of 2.67 and 2.82 mM (2.35 mM Ni: 0.47 mM V), respectively

(Figure 1a,c). However, V treatment of *pmk-1* deletion mutants shows a leftward shift in dose–response curve and significantly lower ($p < 0.001$) $LD_{50}$ at 65.02 mM compared to *wt* worms (Figure 1b), indicative of greater sensitivity to V. Contrastingly, dose–response curves for *pmk-3* mutant strains indicate reduced sensitivity to Ni and metal mixtures as seen by a rightward shift of curves with significantly higher ($p < 0.001$) $LD_{50}$ at 4.03 mM and 4.55 mM (3.79 mM Ni: 0.76 mM V), respectively, compared to *wt* worms (Figure 1a,c). No significant effect is observed for treatment with V. *mpk-1* strains, on the other hand, exhibit reduced sensitivity to all treatment paradigms; we see a rightward shift of curves for exposure to all treatment paradigms, with significantly higher ($p < 0.001$) $LD_{50}$ of 5.03 mM for Ni, 159.50 mM for V, and 6.39 mM (5.33 mM Ni: 1.07 mM V) for mixtures of both metals, compared to *wt* worms (Figure 1a–c). Overall, our results suggest a greater toxicity following Ni exposure relative to V exposure. Most obvious is that deletion of *pmk-3* and *mpk-1* in worms resisted or attenuated Ni and V lethality, with *mpk-1* mutations consistently mitigating lethality across all exposure paradigms. Furthermore, we observed a significant increase in the percentage of developmental delays in all metal exposure scenarios for both *wt* and mutant strains including at concentrations below $LD_{50}$ values (Figure 2). This suggests that even when larvae worms survive low concentrations of metal exposures, development to the adult stage may be impeded.

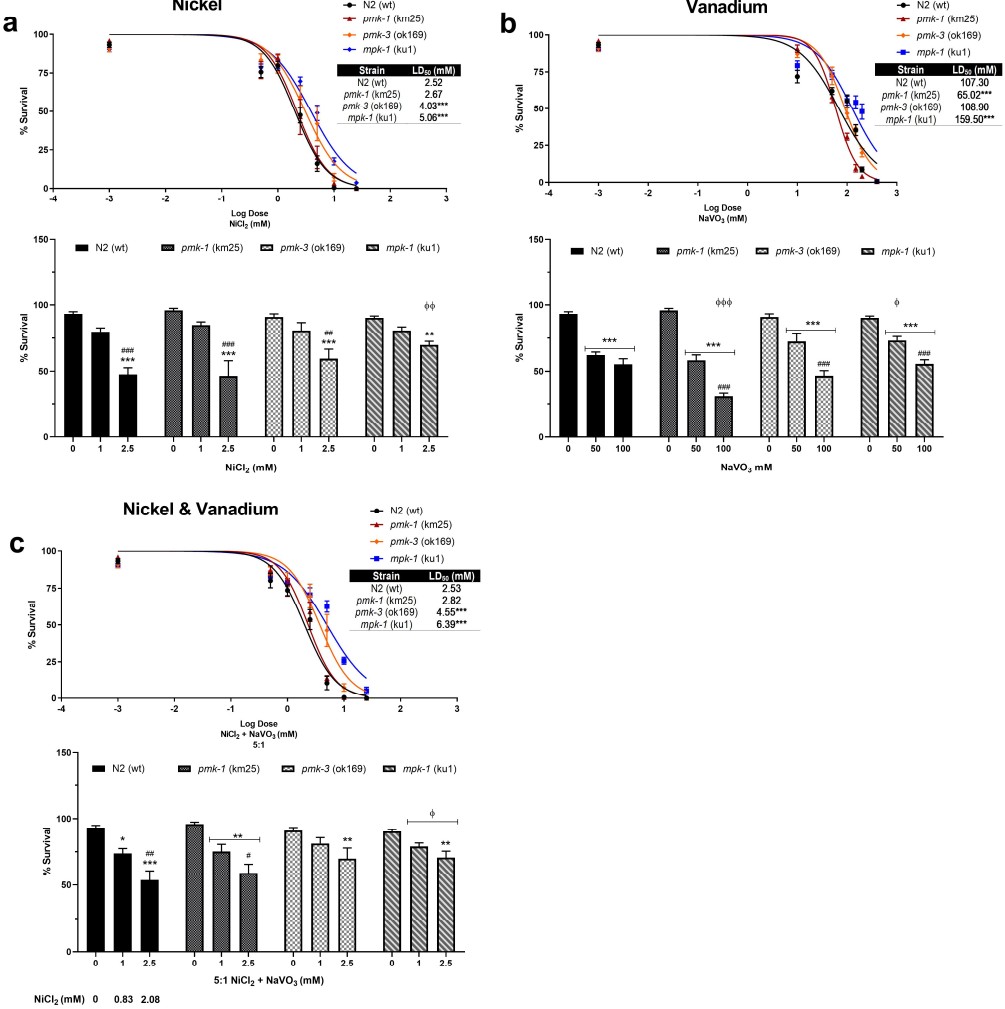

**Figure 1.** *Top*—Dose–responses curves and $LD_{50}$ of *wt* and mutant strains following exposures to Ni (**a**), V (**b**), and metal mixtures (**c**). Sigmoidal dose–response model was used to draw lethality curves and determine $LD_{50}$. * $p < 0.05$, ** $p < 0.01$, *** $p < 0.001$ compared to *wt*. *Bottom*—Percentage survival at selected doses of Ni (**a**), V (**b**), and metal mixtures (**c**) exposure. * $p < 0.05$, ** $p < 0.01$, *** $p < 0.001$

compared to control of the same strain; [#] $p < 0.05$, [##] $p < 0.01$, [###] $p < 0.001$ between low and high doses of the same strain; [φ] $p < 0.05$, [φφ] $p < 0.01$, [φφφ] $p < 0.001$ compared to *wt* of same dose. Data are expressed as mean $\pm$ SEM of three independent experiments. Statistical differences determined with two-way ANOVA followed by Tukey's tests for post hoc.

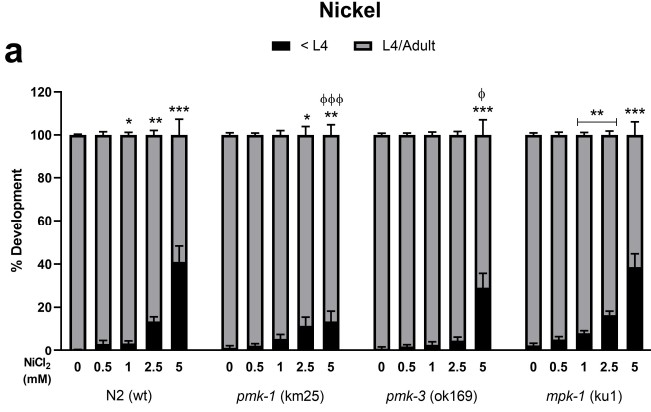

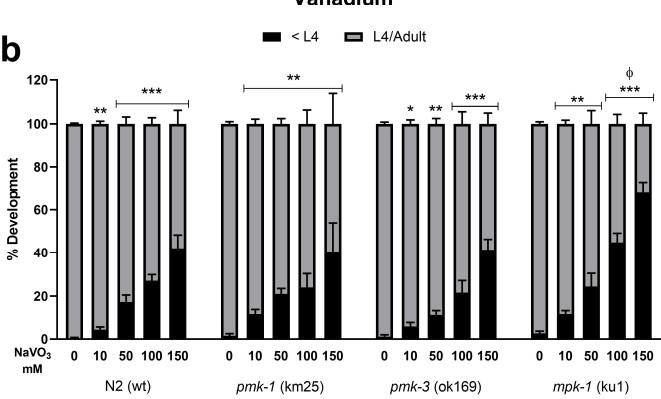

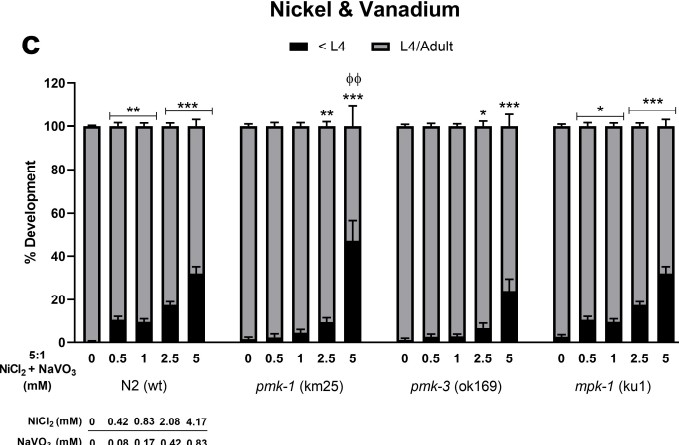

**Figure 2.** Developmental effects on *wt* and mutant *C. elegans* following exposures to Ni (**a**), V (**b**), and metal mixtures (**c**). Data presented as percentage of worms with growth delay (black) and normal development (grey) normalized to the number of surviving worms 48 h post-exposure. * $p < 0.05$, ** $p < 0.01$, *** $p < 0.001$ compared to control of the same strain for worms with growth delay (<L4). [φ] $p < 0.05$, [φφ] $p < 0.01$, [φφφ] $p < 0.001$ compared to *wt* of same dose. Data are expressed as mean $\pm$ SEM of three independent experiments. Statistical differences determined with one-way ANOVA followed by Dunnett's multiple comparison tests.

### 2.2. Metal Bioavailability after Exposures

ICP-OES determination of metal levels showed an increase in Ni and V levels following exposures. Specifically, we observed increased Ni levels following Ni exposure, though not reaching a significant effect for *wt* and *pmk-3* worms; however, *pmk-1* and *mpk-1* worms showed significantly increased ($p < 0.05$) Ni levels after Ni treatment (Figure 3a). Similarly, exposures to mixtures of the metals show a significant increase in Ni levels for *pmk-1* and *mpk-1* at 2.5 mM (2.08 mM Ni: 0.42 mM V) (Figure 3b). On the other hand, V levels were significantly increased ($p < 0.01$) in a dose-dependent pattern in *wt* and all mutant strains following V exposures (Figure 3c). Additionally, a mixture of Ni and V exposure showed an increase in V levels, albeit not reaching a significant effect for any strain except *pmk-3* ($p < 0.05$) (Figure 3d).

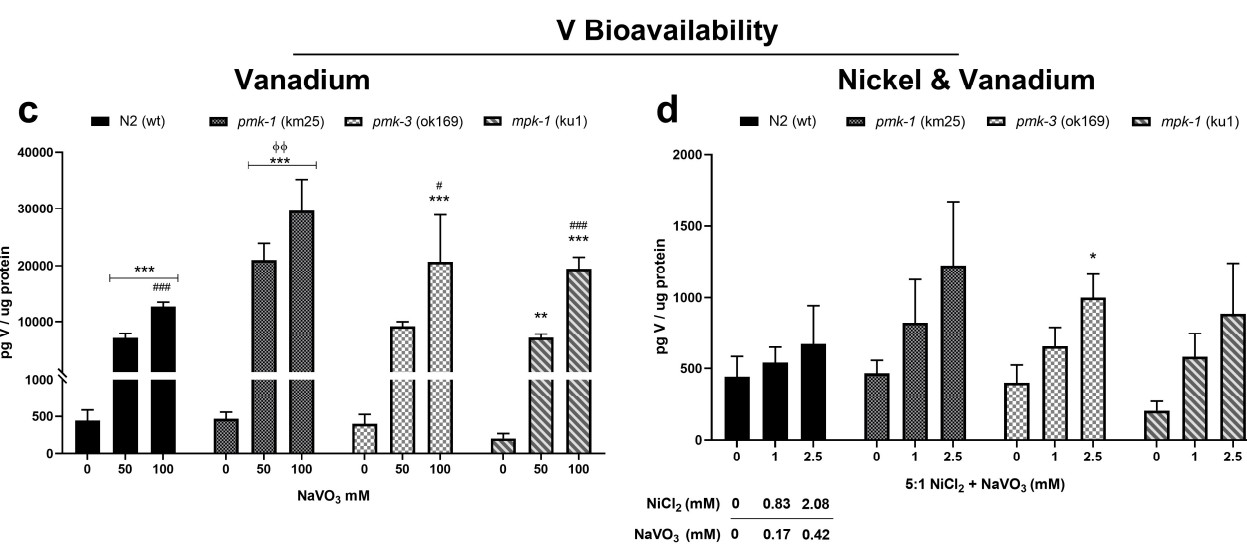

**Figure 3.** ICP-OES measurement of Ni (**a**,**b**) and V (**c**,**d**) in *wt* and mutant *C. elegans* after metal exposures. * $p < 0.05$, ** $p < 0.01$, *** $p < 0.001$ compared to control of the same strain; [#] $p < 0.05$, [###] $p < 0.001$ between low and high doses of the same strain; [ΦΦ] $p < 0.01$ compared to *wt* of same dose. Data are expressed as mean $\pm$ SEM of four independent experiments. Statistical analysis with two-way ANOVA followed by Tukey's tests for post hoc; within strain differences are further confirmed by one-way ANOVA with Tukey's tests.

### 2.3. Changes to Mitochondrial-Derived ROS following Metal Exposures

MitoTracker Red® CM-H$_2$XROS dyes were used to assess increase in mitochondrial-derived ROS and impairment to the mitochondrial membrane potential. Ni and V treatment significantly increased mitochondrial-derived ROS in *wt* worms. Similarly, Ni treatment of *pmk-1* worms also showed significantly increased mitochondrial-derived ROS. On the other hand, Ni treatment of *pmk-3* mutant worms showed no such increase, while *mpk-1* deletion worms showed increased mitochondrial-derived ROS, though the effect was lower compared to *wt* worms (Figure 4a). Contrastingly, following V treatment, *pmk-1*, *pmk-3*, and *mpk-1* worms showed a reduction in mitochondrial-derived ROS, reaching significant effect in *pmk-3*. Also notably, all mutant strains presented significantly lower mitochondrial-derived ROS levels compared to *wt* worms after V treatment (Figure 4b).

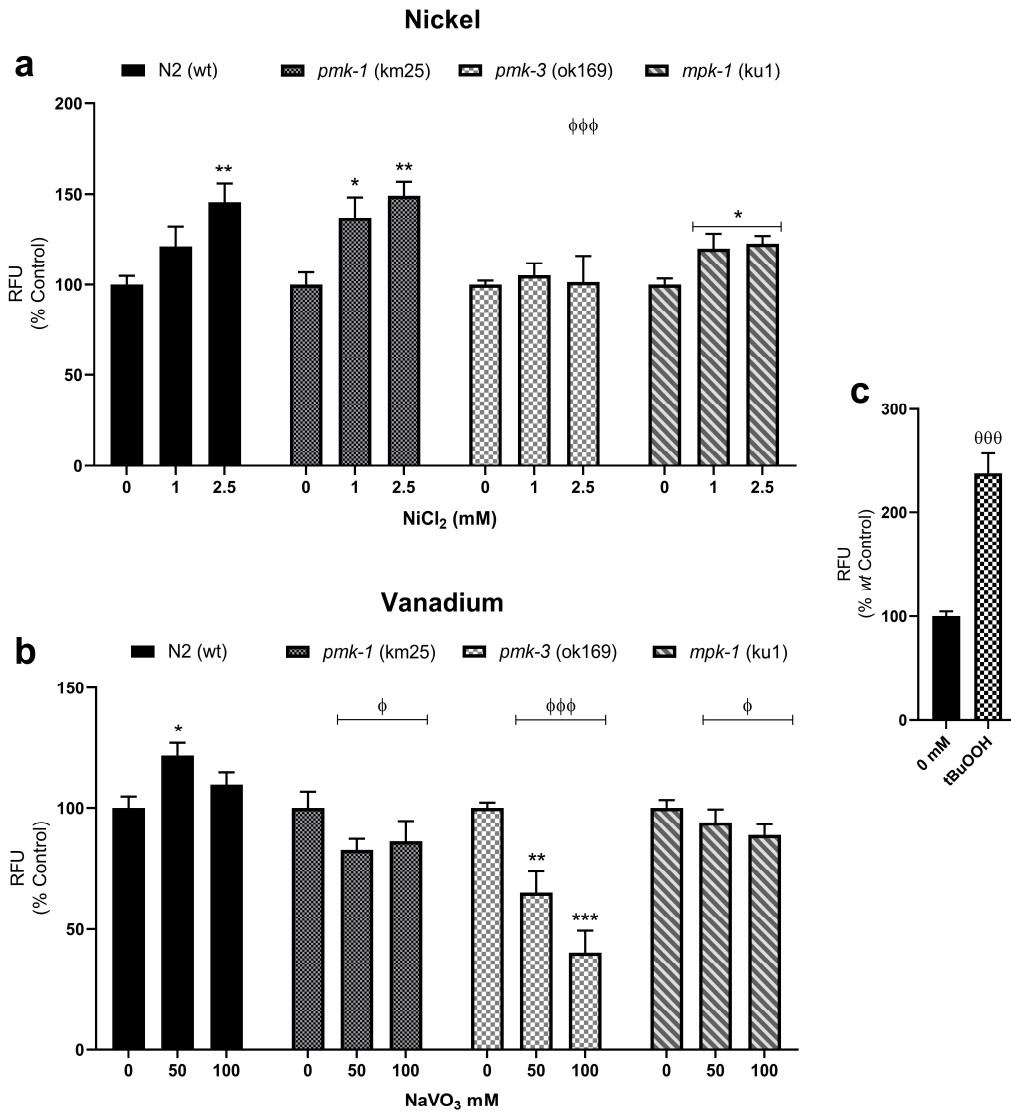

**Figure 4.** Mitochondrial-derived ROS measurements with MitoTracker Red® CM-H$_2$XROS in *wt* and mutant *C. elegans* following exposures to Ni (**a**) and V (**b**). * $p < 0.05$, ** $p < 0.01$, *** $p < 0.001$ compared to control of the same strain; $^\phi$ $p < 0.05$, $^{\phi\phi\phi}$ $p < 0.001$ compared to *wt* of same dose. Data are expressed as mean ± SEM of three independent experiments. Statistical differences determined with one-way ANOVA followed by Tukey's multiple comparison tests. tBuOOH (2 mM) is used as positive control (**c**). $^{\theta\theta\theta}$ $p < 0.001$ *t*-test between experimental control (0 mM) and positive control.

### 2.4. Dopamine and Serotonin Levels after Metal Treatment

LC-MS/MS analysis revealed increased levels of dopamine and serotonin in *wt* worms after Ni treatment, albeit not significant, with no obvious effects after V treatment (Figure 5). Mutant strains showed no obvious effects as well, except for *pmk-1* worms. *pmk-1* worms indicate a reduction in dopamine and serotine levels following Ni treatment, albeit not a significant one (Figure 5a,b). However, following V treatment, dopamine levels in pmk-1 worms significantly increased at 100 mM compared to the control (Figure 5c), while serotonin levels decreased significantly at both 50 and 100 mM (Figure 5d).

**Figure 5.** LC-MS/MS measurements of dopamine and serotonin levels after Ni (**a**,**b**) and V (**c**,**d**) exposures in *wt* and mutant *C. elegans*. * $p < 0.05$, ** $p < 0.01$ compared to control of the same strain; $^{\Phi}$ $p < 0.05$, $^{\Phi\Phi}$ $p < 0.01$ compared to *wt* control. $^{\nabla}$ $p < 0.05$, $^{\nabla\nabla}$ $p < 0.01$, $^{\nabla\nabla\nabla}$ $p < 0.001$ compared to *pmk-1* of same dose. Data are expressed as mean ± SEM of four independent experiments. Statistical analysis with two-way ANOVA followed by Tukey's tests for post hoc.

### 3. Discussion

Here, we attempt to elucidate the role of p38- and ERK-MAPK in neurotoxicity after early life exposures to Ni and V using the *C. elegans* model. The current result supports our earlier reports where we have shown that Ni induces lethality in developing *C. elegans* at very low concentrations [16]. Other studies have also corroborated that Ni exposure reduces *C. elegans* survival from larval stages to adults [28–30]. Similarly, other studies have established Ni toxicities in both human and vertebrate animal models, particularly its neurotoxic effects [31,32]. Likewise, toxicity following overexposure to V has also been documented, although knowledge about V toxicity is relatively scarce. V overexposure leads to morphological and functional injuries to peripheral organs including the liver, kidney, spleen, and to the central nervous systems, particularly the brain [11,33–35]. Furthermore, the possibility of V derivatives, such as oxovanadium complexes, for potential therapeutic uses have been touted, hence triggering the need to understand potential side

effects [34,36], in addition to its toxicity due to environmental overexposure. Hence, it is noteworthy that in the present study, we provide the first report to use the *C. elegans* model for V toxicity, establishing an LD$_{50}$ for acute V exposure during development for *C. elegans*. We also observe here that a much higher concentration of V is needed to trigger lethality relative to Ni exposures. In the present study, we obtained an LD$_{50}$ of 2.52 mM for Ni exposure in wild-type worms relative to 107.30 mM obtained for V. Our previous study has shown a similarly low concentration for toxicity [16]. Overall, we can speculate that Ni poses a greater toxicity in this model compared to V.

Furthermore, we also report toxicity of mixes of Ni and V. Ni and V are the dominant metal compositions in crude oil; however, experimental studies examining combined effects are lacking [37–39]. This is hugely relevant to understanding the full impact of metal toxicity considering synergistic effects of metals, which could potentiate or alleviate effects on organisms. In most of Nigeria's crude oil blends, there is a greater ratio of Ni to V [7,8]; hence, our 5:1 ratio selection in the present study. Given the greater content of Ni, it is possible that observed toxicities are driven largely by Ni levels in the mixtures. This can be expected considering the present data which have shown a much lower concentration of Ni is needed to trigger toxicities in the worms. In the environment, metals exist as mixtures. Multiple metals may enter the body at the same time through food, water, or air [40,41]. Studies that have evaluated the toxicity of metal mixes show that effects of mixes are difficult to predict and there can be conflicting interactions across various experiments [40]. However, given that realistically, metals co-exist, and their interactions could modulate their impact on an organism, it is essential to study the impact of metal mixes with a focus on protocol standardization across different experimental settings to better delineate metal neurotoxicities.

Additionally, we also document developmental delays (growth from larval to adult) in the *C. elegans* for both Ni and V treatments. Previous studies have indicated genomic and/or physiological damage to nematodes gametes following Ni exposure [29,42]. Specifically, a recent study also observed delayed growth in worms as indicated by smaller body length 48 h in apparently normal wild-type worms after Ni exposures, which is supported by our present data [30]. Similarly, the impact of V on reproductive indices and development has been severally documented in rodent studies. Available reports indicate functional and morphological damage to both male and female gametes as well as negative indices on litter weight, viable foetuses, and incidence of foetal anomalies following V exposures [43]. Though there is a dearth of prior reports of V on worm development, a previous study utilizing the invertebrate sea urchins showed that V exposure caused obvious delay in embryo development, and drastic reduction in total skeletal mass [35]. It is thus possible that reduced body length caused by Ni and V observed in the present study could be due to a reduction in muscle mass. Though, the *C. elegans* have a fixed number of somatic cells, a greater percentage of growth that happens during development occurs after cessation of cell division; hence, volumetric growth in *C. elegans* is largely due to increased sizes of some or all cells [44]. We can thus speculate that metals may impact mechanisms in the worm that regulate growth (increase in cell size), such as the DBL-1/TGF-β pathway, which regulates post-embryonic growth in worms [44,45]. Interestingly, it has been reported that Ni could activate the TGF-β signalling in the epithelial–mesenchymal transition [46]. Nevertheless, further studies on these potential mechanisms are warranted.

Importantly, we show that both Ni and V developmental toxicity in worms can be modulated by p38- and ERK-MAPK signalling pathways. Previous experimental studies have shown that Ni compounds induce inflammatory and apoptotic changes that are influenced by these pathways. NiCl$_2$ treatment of bone marrow-derived macrophages (BMDMs) resulted in increased phosphorylation of p38 MAPK and ERK proteins, which was further linked to upregulation of pro-inflammatory cytokines including TNF-α, IL-1β, IL-6, IL-8, and INF-β [47]. Similarly, treatment with nickel subsulfide (Ni$_3$S$_2$) was shown to induce apoptosis in human bronchial epithelial cells (BEAS-2B) via an increase in p38 expression due to the activation of upstream Akt and ASK1 proteins [48]. Likewise, BEAS-

2B treatment with nickel oxide nanoparticles also resulted in apoptosis and inflammatory response via the activation of p38 proteins [49]. Given the involvement of these pathways, studies have shown their inhibition could mitigate the toxic effects of Ni. A study reported that inhibition of ERK1/2, but not p38 MAPK, suppressed nickel sulfate ($NiSO_4$)-induced TNF-$\alpha$ release [50]. However, another study has shown that epigallocatechin-3-gallate (EGCG) attenuates toxicity induced by nickel nanoparticles in the mouse epidermal cell line (JB6 cell) via the inhibition of both p38 and ERK1/2 proteins [51]. Furthermore, one available study using the *C. elegans* model [52] reported that loss-of-function p38 MAPK genes, *pmk-1* and *pmk-3*, suppressed Ni-induced germline apoptosis following exposure to $NiSO_4$, while the ERK gene, *mpk-1*, had no effects. Here, we also show that loss-of-function of *pmk-3* mitigates Ni-induced lethality. Nevertheless, in contrast to the aforementioned study, we show that loss of *pmk-1* was not affected, while *mpk-1* loss-of-function mutants attenuated lethality following Ni exposure. Similarly, previous reports also show that V exposure influences p38- and ERK-MAPK in both mammalian and non-mammalian-derived cells. Ingram and colleagues reported a time-dependent activation of p38 MAPK and ERK1/2 proteins after Vanadium pentoxide ($V_2O_5$) treatment in human lung fibroblast. Furthermore, pretreatment with respective inhibitors of these proteins blocked V-induced activation [53]. Likewise, Chien et al. also supported that V is a potent activator of p38 MAPK and ERK1/2 in A549 human lung carcinoma cell lines [54]. Additionally, utilizing oviduct magnum epithelial cells from hens, other authors showed that treatment with ammonium metavanadate significantly increased the protein level of p-p38 MAPK and p-ERK1/2 [55]. However, an earlier study utilizing cerebellar granule progenitors in rats showed that though sodium metavanadate treatment produced a transient activation of ERK1/2, it had little effect on its expression. Contrastingly, the authors reported no significant alterations to either expression or activation of p38 MAPK [56]. Here we show that loss-of-function of the ERK gene (*mpk-1*) resulted in reduced sensitivity after V treatment, thus suggesting its importance in influencing V toxicity. Contrastingly, our data suggest that loss-of-function of p38 MAPK genes (*pmk-1* and *pmk-3*) in worms may either not influence V toxicity or significantly worsen it.

Previous studies have indicated that the generation of ROS may play important roles in both Ni and V toxicities. Both metals have been reported to trigger ROS including peroxides, superoxide, and hydroxyl radical, amongst others. The ROS generation by Ni and V can be both a consequence as well as a cause of mitochondrial dysfunctions, which is also established in toxicities of both metals [31,32,34,57]. Furthermore, both metals have been reported to induce mitochondrial production of ROS [47,58]. Similarly, our results support the generation of mitochondrial-derived ROS and impairment to the mitochondrial membrane potential following Ni and V treatment in *wt* worms. However, while loss of *pmk-1* maintained increased mitochondrial-derived ROS after Ni treatment, *pmk-3* worms showed no changes but rather appeared to lower ROS compared to *wt* worms. On the other hand, *mpk-1* worms also show increased ROS; however, to a much lesser extent compared to *wt* and *pmk-1* worms. This supports our lethality data, which suggest that *pmk-1* mutation may have an impact on Ni toxicity. In contrast, following V exposure, loss of *pmk-1*, *pmk-3*, and *mpk-1* resulted in lowered mitochondrial ROS compared to *wt* worms. In particular, *pmk-3* mutants showed a significant dose-dependent reduction in mitochondrial ROS. It is unclear why this is the response; however, we suggest that inhibiting the p38 and ERK pathways may prevent the generation of mitochondrial-derived ROS after V exposures.

Previous studies have reported alterations to levels of neurotransmitters including dopamine and serotonin. Several rodent experimental studies have shown that Ni (depending on concentrations) can both increase and decrease dopamine levels in the cortex and basal ganglia, as well as reduce serotonin levels by suppression of its receptor gene expression [59,60]. Similarly, V exposure has been linked to lowered levels of tyrosine hydroxylase and dopamine [11]. Here, we see no significant impact on dopamine and serotonin levels in *wt* worms after Ni and V treatments. However, here we consistently

observe that the loss of *pmk-1* alters levels of dopamine and serotonin, which may be due to the greater metal uptake of this deletion mutant as our data suggest.

## 4. Materials and Methods

### 4.1. C. elegans Strains and Maintenance

*C. elegans* were grown at 20 °C on nematode growth media (NGM) or 8-fold peptone (8P) nematode media with bacterial diet of *Escherichia coli* strains OP50 or NA22, respectively. The following strains were used: N2 *wild-type (wt)*, KU25 [*pmk-1(km25)*] BS3383 [*pmk-3; (ok169)*], MH37 [*mpk-1(ku1)*]. All strains were obtained from the *Caenorhabditis* Genetics Center (CGC), USA. Synchronization of worm population was performed using bleaching method (in 1% NaOCl and 0.25 M NaOH solution), and eggs were separated by floating in 30% sucrose gradient.

### 4.2. Preparation of Stock Solutions

Ni was administered as nickel chloride hexahydrate ($NiCl_2·6H_2O$) and V as sodium metavanadate ($NaVO_3$). Both chemicals were procured from Sigma-Aldrich, Darmstadt, Germany. Stock solutions of 1 M $NiCl_2·6H_2O$ and 1M $NaVO_3$ were prepared, respectively, in ultrapure MilliQ water.

Ni and V solutions were diluted from stock in 85 mM NaCl for varying concentrations of $NiCl_2$, $NaVO_3$, and $NiCl_2$ + $NaVO_3$ and were used for subsequent experiments. $NiCl_2$ + $NaVO_3$ solutions were prepared at a ratio of 5:1 of Ni to V based on environmentally relevant ratios of Nigeria's crude oil blends [7,8].

### 4.3. Survival and Developmental Assays

Acute exposure and survival assay were performed as previously described [16,61]. In brief, 2500 synchronized larvae stage 1 (L1) *wild-type* and mutant (*pmk-1, pmk-3,* and *mpk-1*) worms were treated for 1 h with varying concentrations of $NiCl_2$, $NaVO_3$, or $NiCl_2$ + $NaVO_3$. Treatment was performed by rotating worms in 1.5 mL tubes containing 500 μL solution. For these assays, concentrations of $NiCl_2$ range from 0 to 25 mM (0, 0.5, 1, 2.5, 5, 10, and 25 mM), concentrations of $NiCl_2$ range from 0 to 200 mM (0, 10, 50, 100, 150, and 200 mM), and metal mixtures at a ratio of 5:1 of $NiCl_2$ + $NaVO_3$ as aforementioned. To perform the survival assays, 20–40 worms were transferred to OP50-seeded NGM plates in triplicates, after metal treatments. The total number of worms that survived after 48 h was counted and scored as a percentage of the initial number of worms plated. For developmental assay, their developmental stage was recorded 48 h after treatment, and worms failing to reach the L4 stage were counted. The percentage of worms with developmental delay was calculated as ([number of worms failing to reach L4/number of worms reaching L4] × 100) [62].

### 4.4. Metal Bioavailability

Evaluation of Ni and V bioavailability was performed via inductively coupled plasma—optical emission spectrometry (ICP-OES (Spectro, Kleve, Germany) as previously described [63]. Seventy thousand L1 worms were subject to acute metal exposure as described above. Following metal treatments, worms were pelletized by centrifugation and remaining *E. coli* was removed by four washing steps with 85 mM NaCl + 0.01% Tween. Three freeze–thaw cycles and sonication (UP100H ultrasonic processor (Hielscher, Teltow, Germany), 3 × 20 s, 100% amplitude, cycle: 1) allowed complete rupture of the worms' cuticle. Samples were dried and acid-assisted digested in a 1:1 mixture of 65% nitric acid ($HNO_3$) and 30% hydrogen peroxide ($H_2O_2$) at 95 °C. Ashes were diluted in a 2% $HNO_3$ solution and measured analytically using the following ICP-OES method parameters: plasma power: 1400 W, cooling gas flow: 12 L/min, auxiliary gas flow: 1 L/min, nebulizer gas flow: 1 L/min, nebulizer type: MicroMist®. Evaluated wavelengths were 311.071 nm for V and 232.003 nm for Ni. Metal amounts were validated by the measurement of acid-assisted digested reference material BCR®-274 (Single Cell Protein, Institute for Reference Materials

and Measurement of the European Commission, Geel, Belgium). Ni and V amounts were normalized to protein content determined by BCA assay (bicinchoninic acid assay). A large number of worms are required to obtain an adequate amount of protein content, particularly when working with L1 worms. Though 50,000 worms are sufficient for most analysis as with MitoTracker and neurotransmitter measurements described below, for metal measurements, we observed that to obtain accurate reads on the ICP-OES from L1 stage worms, we needed to increase the number of worms to 70,000.

### 4.5. MitoTracker Dyes and Fluorescence Quantification

MitoTracker Red® CM-H$_2$XROS dyes (Thermo Fisher Scientific, Waltham, MA, USA) were used to assess mitochondrial membrane potential- and mitochondrial-derived ROS [64]. First, 50,000 L1 worms were incubated in 2.5 mM MitoTracker Red in the dark for 2 h. After which, worms were washed four times in 85 mM NaCl, and subjected to metal treatments for 1 h. Simultaneously, N2 worms were treated with 2 mM tert-butyl hydroperoxide (tBuOOH) for positive control. After metal treatments, worms were washed three times in 85 mM NaCl containing 0.01% Tween 20 and transferred to OP50-seeded NGM plates for 30 min to allow for excretion of excess dyes. Afterwards, worms were washed off plates into 1.5 mL tubes, and transferred to 96-well plates in triplicates of 100 μL volume. The fluorescence was monitored via a microplate reader (Tecan Infinite M200 Pro, Tecan Group Ltd Männedorf, Switzerland) at excitation 560 nm/emission 599 nm.

### 4.6. Neurotransmitter Measurements

We evaluated dopamine (DA) and serotonin (SRT) levels via LC-MS/MS as previously described [65]. First, 50,000 L1 worms were subjected to acute metal exposure as described above. Worms were washed three times in 85 mM NaCl containing 0.01% Tween 20 and pelletized by centrifugation at 4660 g. Then, 100 μL extraction buffer (2 mM sodium thiosulfate, 200 mM perchloric acid and 25 nM each of deuterated analogues of DA (2-(3,4-dihydroxyphenyl)ethyl-1,1,2,2-d$_4$-amine (DA-d$_4$), CDN Isotopes) and SRT (serotonin-α,α,β,β-d$_4$ creatinine sulfate complex (SRT-d$_4$), CDN Isotopes)) were added on each sample, following 4x freeze–thaw cycles (1 min liquid nitrogen and 1 min 37 °C). The samples were homogenized by sonication following centrifugation at 16,060 g and were transferred into vials. An aliquot was reserved for protein quantification via BCA assay. The analysis of dopamine and serotonin was conducted on an Agilent HPLC System (Agilent Infinity II, Agilent, Santa Clara, CA, USA) interfaced with a Sciex triple quadrupole mass spectrometer (Sciex, Framingham, MA, USA) with an electrospray ion source operating in positive ion mode. Terms for chromatographic separation as well as ion source parameters were obtained previously [65]. The MRM transitions chosen for quantification were $m/z$ 154 > 91 for DA, $m/z$ 158 > 95 for DA-d$_4$, $m/z$ 177 > 160 for SRT, and $m/z$ 181 > 164 for SRT-d$_4$. Values are expressed as ng DA or ng SRT per mg protein.

### 4.7. Statistics

Each experiment was repeated for 3 independent replicates. Data were statistically analysed on GraphPad Prism 8 software (GraphPad Inc, La Jolla, CA, USA). Dose–response lethality curves and LD$_{50}$ determination were performed using a sigmoidal dose–response model with a top constraint at 100%. Statistical differences between the metal concentrations in the same strains (genotype) were determined with one-way randomized ANOVA followed by post hoc tests. Furthermore, interaction between treatments and genotypes were analysed with two-way ANOVA followed by post hoc tests. Where appropriate, Student's t-test was used to confirm significant differences between two groups. Values of $p < 0.05$ will be considered statistically significant.

## 5. Conclusions

In conclusion, here we demonstrate the developmental toxicity of Ni and V in *C. elegans*, developing for the first time a *C. elegans* model for V toxicity testing. We show that both Ni

and V induce lethality, developmental delays, and a mitochondrial-derived ROS increase in worms; however, V requires a much higher concentration. Also, we show that the loss of *pmk-1*, *pmk-3,* and *mpk-1* worm homologues to mammalian p38- and ERK-MAPK genes influences the *C. elegans* response to Ni and V toxicities. Overall, our results suggest the p38- and ERK-MAPK signalling pathways may modulate Ni and V neurodevelopmental toxicity via influence on mitochondrial health, metal level dynamics, and neurotransmitter regulation. Further studies will unravel behavioural phenotypes in these mutant strains after metal exposures and measure target genes of *pmk-1*, *pmk-3,* and *mpk-1*.

**Author Contributions:** Conceptualization, O.M.I.; Formal analysis, O.M.I., A.-K.W., V.M., O.K.I. and J.B.; Funding acquisition, O.M.I. and J.B.; Investigation, O.M.I., A.-K.W. and V.M.; Methodology, O.M.I., A.-K.W., V.M. and J.B.; Project administration, O.M.I. and J.B.; Writing—original draft, O.M.I.; Writing—review and editing, A.-K.W., V.M., O.K.I. and J.B. All authors have read and agreed to the published version of the manuscript.

**Funding:** This work was funded by the Alexander von Humboldt (AvH) Georg Forster Research Fellowship for Experienced Researchers (NGA-1216466-GF-E) awarded to OMI. Further, OKI acknowledges the AvH Georg Forster Research Fellowship for Postdoctoral Researchers (NGA-1218847-GF-P). Additionally, JB acknowledges the German Research Foundation (DFG) funding (BO4103/4-2).

**Institutional Review Board Statement:** Not applicable.

**Informed Consent Statement:** Not applicable.

**Data Availability Statement:** Data supporting the study can be made available by the authors upon request.

**Acknowledgments:** *C. elegans* strains were obtained from the *Caenorhabditis* Genetics Center (CGC), which is supported by the National Institutes of Health—Office of Research Infrastructure Programs (P40 OD010440).

**Conflicts of Interest:** The authors declare no conflicts of interest.

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
