# Peer review of "p38- and ERK-MAPK Signalling Modulate Developmental Neurotoxicity of Nickel and Vanadium in the Caenorhabditis elegans Model"

_2813-3757, doi:10.3390/kinasesphosphatases2010003_

Round 1
Reviewer 1 Report
Comments and Suggestions for Authors
Key words should be different from the title.
The conclusions in the abstract should be in line with the general conclusions on page 384. For example: however, V requires a much higher concentration.
In the introduction, you should talk more about the Caenorhabditis elegans model and its applications.
The figures would be better appreciated by the reader if they were enlarged. For example, in figure 1, it is difficult to visualise the lines, as they are too close together, and it would be better to just leave the bar graphs.
The work is very well discussed and deserves to be published.
Reviewer 2 Report
Comments and Suggestions for Authors
The manuscript deals with the neurotoxicity induced by two heavy metal and uses a valuable model organism. Overall, it is interesting, however minor revisions are needed - with necessary clarifications in the materials and methods section - and an extensive revision of the discussion is required: the data are not discussed adequately.
Here some minor errors to be fixed:
Introduction:
I would mention in which organisms Ni and V-induced neurotoxicity has been seen.
Materials and Methods:
Paragraph numbers are missing.
Lines 89-95: I would use a past tense rather than a present tense, considering that you are explaining a procedure.
Lines 101-102: what are these "various concentrations" you talk about? Is there an explanation on the choice of concentrations used?
107-116: standardize verb tenses, preferably using the past tense.
Line 108: Were 2.500 larvae used for each treatment? Wt, pmk-1, pmk-3, and mpk-1 ? 2500x4 is 10.000. Is this the total number? In line 119 a different number of the total is reported.
Line 119: there is an error, the sentence starts with a reference [29].
Line 119: it is not clear why 70.000 larvae were treated. There is no correspondence with the number above. Clarify.
Line 137 and 150: 50.000 worms used for MitoTracker and Neurotransmitter measurements are different from those used by the previous treatments? How were they divided into the various treatments? Explain
Results:
Paragraph numbers are missing.
Lines 196-203: these assumptions should be brought into the discussion section. The results simply report what was observed from the experiments.
Figure 1: The graphs are of poor quality. The writing is grainy. It would be better to space the graphs apart from each other and insert the name of the treatment above the relevant graphs, to make the reader's understanding more immediate: the writing on the axes is quite small. I would repeat the same thing in all the graphs.
Discussion: I find that the discussion is mainly focused on studies carried out by other authors (with characteristics similar to a review) with little reference to the data obtained from the present work. The study aims to demonstrate the toxicity of Ni and V via their influence on mitochondrial health, metal level dynamics and neurotransmitter regulation, but no speculation is made or possible explanations are given for this. I suggest an intense review of the discussion, highlighting your results, discussing and using other literature data to support it.
Comments on the Quality of English LanguageCan be improved. There is a continuous switch between present and past time, it must be uniform throughout the manuscript.
Round 2
Reviewer 2 Report
Comments and Suggestions for Authors
Dear Authors,
the manuscript has been sufficiently improved, however, I believe it is necessary to return to some points that, in the previous round of revisions, you decided to overlook.
1) Paragraph numbers are still missing: for order and logic it is necessary to insert the numbering of the subparagraphs (e.g.: 2.1 C. elegans strains and maintenance; 2.2 Preparation of stock solutions etc for all the sections). Line numbering is not visible once the manuscript is published, and opening the latest manuscripts published by the paper you will realise that the correct numbering of subparagraph has been inserted. Therefore, apply the required modifications.
2) In Materials and methods, you used different numbers of animals for the various experiments. It should be better explained - in the text, therefore readable to all future readers- and not just as a response to the reviewer.
3) Graphs resolution might improve during production by the editorial office, but the confusion generated by a multitude of data remains. As previously suggested, it would be better to make it clearer, as the header of each graph, which treatment is the subject of each graph.
